# Effect of Vaccination against *Leptospira* on Shelter Asymptomatic Dogs Following a Long-Term Study

**DOI:** 10.3390/ani12141788

**Published:** 2022-07-12

**Authors:** Ricardo Sant’Anna da Costa, Maria Isabel N. Di Azevedo, Ana Luiza dos Santos Baptista Borges, Luíza Aymée, Gabriel Martins, Walter Lilenbaum

**Affiliations:** 1Laboratory of Veterinary Bacteriology, Department of Microbiology and Parasitology, Universidade Federal Fluminense, Niterói 24210-130, Brazil; ricardo-santanna@hotmail.com (R.S.d.C.); isabeldiazevedo@gmail.com (M.I.N.D.A.); analuizaborges@id.uff.br (A.L.d.S.B.B.); luizaaymeeps@gmail.com (L.A.); 2Centro Universitário Arthur Sá Earp Neto, Petrópolis 25680-120, Brazil; gmartins@id.uff.br

**Keywords:** canine leptospirosis, sequencing, commercial vaccine, silent infection, leptospiral infection, subclinical infection, animal host

## Abstract

**Simple Summary:**

Dogs are known as hosts of *Leptospira interrogans* and can become asymptomatic carriers of leptospires in the urine, spreading this bacterium to the environment, especially in endemic areas, characterizing a serious public health problem. In this context, vaccination of dogs against leptospirosis is of paramount importance for long-term protection against renal carrier status. A total of 118 dogs were studied for 365 days, separated into Group A (vaccinated, *n* = 94) and Group B (non-vaccinated, *n* = 24). Group A was subdivided into three groups: A1 with 32 dogs immunized with the vaccine #1; A2 by 32 dogs with #2; and A3 30 dogs with #3. Serology (MAT and IgG-ELISA) and urinary PCR were conducted. Seroreactivity increased at D15 post-vaccination and, regardless of vaccine brand, remained high up to D180, with antibody switch to IgG after D30. A total of 46.8% of animals from Group A were PCR-positive at least once, in contrast to 75% in Group B, regardless of vaccine brand.

**Abstract:**

(1) Background: Vaccination of dogs against leptospirosis is of paramount importance, as they ideally must provide not only long-term protection, but also against the renal carrier state of leptospires. This study assessed the post-vaccine humoral response against *Leptospira* in naturally exposed dogs and effects on renal carrier status. (2) Methods: A total of 118 dogs were studied for 365 days, separated into Group A (vaccinated, *n* = 94) and Group B (non-vaccinated, *n* = 24). Group A was subdivided into three groups: A1 with 32 dogs immunized with the vaccine #1; A2 by 32 dogs with #2; and A3 30 dogs with #3. Serology (MAT and IgG-ELISA) and urinary PCR were conducted. (3) Results: Seroreactivity increased at D15 post-vaccination and, regardless of vaccine brand, remained high up to D180, with antibody switch to IgG after D30. A total of 46.8% of animals from Group A were PCR-positive at least once, in contrast to 75% in Group B, regardless of vaccine brand (*p* < 0.05; OR: 0.3). (4) Conclusions: All commercial vaccines succeeded at eliciting a long-term IgG-based response and were partially effective at protecting against kidney infection.

## 1. Introduction

Canine leptospirosis is described worldwide [1] and was recognized as a disease of dogs in 1899 [2]. Due to the high level of exposure to pathogenic leptospires, dogs are highly susceptible to infection, both in endemic and non-endemic areas, becoming an important maintenance host of adapted strains and a sentinel for human infection for incidental ones [3]. Leptospiral infection in dogs may range from subclinical disease, with mild and transient signs, or develop dysfunction of multiple organs leading to fatal outcomes [4,5] In addition, leptospires affect the kidneys and are maintained by carrier animals, which spread the bacteria in the urine, remaining in the environment [6]. The role of dogs as asymptomatic carriers has been increasingly studied, as it is known that they can also act as a silent source of infection, representing a serious public health concern [7].

Dogs can shed a myriad of serovars from different species for long periods [8], and vaccination is considered to be the main tool for prevention of the clinical disease [2]. Licensed vaccines against canine leptospirosis are inactivated whole bacteria (bacterins) or purified antigens from the cell wall of cultured bacteria. Protection against leptospirosis through inactivated vaccines is strongly restricted to homologous serovar [9], and directed towards *Leptospira* lipopolysaccharide (LPS) [10]. Despite that, cell-mediated immunity, although not extensively studied, seems to also be important for this immunity. Traditional vaccines used in dogs used to be bivalent, composed of serovars Icterohaemorrhagiae and Canicola [11]; however, other serogroups, as Pomona and Grippotyphosa, are also included [3]. It is largely known that strains of Icterohaemorrhagiae, mainly Copenhageni L1-130, are the most frequent agents of canine and human leptospirosis in Brazil [12].

Sterile immunity is controversial in vaccinated dogs against leptospirosis. While renal carrier protection has been described after vaccination [4], it has also been suggested that vaccines do not prevent from a chronic carrier state in dogs and that the level of protection may vary between vaccines [13]. In addition, the majority of these studies were conducted in controlled experimental conditions and assessed a short to medium-term response. Thus, the real effect of vaccines on the kidney colonization on naturally exposed dogs from an endemic area with high infection pressure remains to be elucidated. Therefore, the objective of the present study was to assess the post-vaccinal humoral responses and effects on leptospiral shedding in urine in naturally exposed dogs throughout one year.

## 2. Material and Methods

This study was approved by the Animal Use Ethics Committee of the Federal Fluminense University under number n° 3778190419.

### 2.1. Dogs and Vaccines

For that study 118 adult dogs were included, male or females, kept in a dog shelter in the metropolitan region of Rio de Janeiro, Brazil. That region is known to be endemic for canine and human leptospirosis [6]. Inclusion criteria were as follows: not vaccinated for leptospirosis for at least 12 months, no apparent symptoms of acute clinical disease, regular hematobiochemical condition (blood count, serum ALT and AST activity, serum creatinine and urea concentrations), seronegative for leptospirosis (Microagglutination Test—MAT), and negative at urinary PCR. Dogs were randomly divided into Group A (*n* = 94), vaccinated for leptospirosis and subdivided into three subgroups: Group A1 consisting of 32 dogs immunized with vaccine 1, Group A2 with 32 dogs immunized with vaccine 2, and Group A3 composed of 30 dogs immunized with vaccine 3. Group A dogs were vaccinated with only one dose of vaccine, with no booster. In addition, 24 animals were not vaccinated, representing the control Group B. All animals were kept together throughout the experiment and subjected to the same environmental conditions. Inspection was performed by the same veterinarian on all visits. He is an experienced certified professional. He followed the dogs throughout the study, and he has been voluntarily in charge of the shelter for at least 15 years. The following commercial vaccines were studied, all of them bacterins: Vaccine 1—Vanguard plus V10 (Zoetis^®^, São Paulo, Brazil), prepared, according to the manufacturer, with serovars Canicola, Icterohaemorrhagiae, Grippotyphosa and Pomona; Vaccine 2—Canine 1-DHPPI + L (Nobivac^®^, São Paulo, Brazil) prepared, according to the manufacturer, with serovars Canicola and Icterohaemorrhagiae/Copenhageni; and Vaccine 3—Recombitek^®^ C6/CV (Merial^®^, Campinas, Brazil), prepared, according to the manufacturer, with serovars Canicola and Icterohaemorrhagiae.

### 2.2. Sampling

All dogs were monitored for 365 days, in which blood and urine samples were collected every 30 days, and the animals were physically examined, always by the same veterinarian, checking temperature, hydration, mucosa color, perfusion time capillary and cardiorespiratory auscultation. Blood samples were collected, and serum was used for microscopic agglutination test (MAT) and IgG-ELISA. MAT was performed at D15, D30, D60, D90, D120, D150, D180, D210, D270, and D365. IgG-ELISA at D15, D30, D60, and D90. At days D0, D90, D180, D270, and D365 urine samples were collected by catheterization using a sterile urethral tube for *lipL*32-PCR. Briefly, 5 mL of urine was collected and an aliquot of 1 mL was transferred to microtubes containing 100 mL of 10× PBS. Samples were kept frozen at −20 °C until molecular analysis.

### 2.3. Serology

#### 2.3.1. Microscopic Agglutination Test (MAT)

For the detection of anti-*Leptospira* antibodies, a microscopic agglutination test (MAT) was conducted according to international recommendations using a panel including eight serovars representing seven serogroups. The antigens used were *Leptospira interrogans* serovars Autumnalis (Akiyami A), Bratislava (Jez-Bratislava), Bataviae (VanTienen), Canicola (Hond Utrecht IV), Grippotyphosa (Moska V), Icterohaemorrhagiae (RGA), Copenhageni (M 20), and Pomona (Pomona). The reaction titer with 50% of agglutinated leptospires corresponded to the reciprocal of the highest dilution of serum, and animals were considered as reactive when the titer was ≥100 [14].

#### 2.3.2. IgG-ELISA

Indirect in-house IgG-ELISA was used and performed as described [15], the recombinant protein lipL32 (rLipL32) was used as antigen, kindly provided by Prof. Odir Dellagostin, UFPel, Brazil. Briefly, polystyrene ELISA microtiter plates (Immulon Microtiter, Thermo Scientific, Waltham, MA, USA) were sensitized with rLipL32 diluted in sodium carbonate-bicarbonate solution with a pH between 9.6 and 9.8. An optimal concentration of 100 ng/well of rLipL32 was used and serum samples were diluted 1:400. The plates were kept overnight at 4 °C. After sensitization, plates were washed with PBS T wash solution and blocked with PBS T solution with 5% skimmed milk powder. The serum samples were thawed, homogenized, and diluted 1:60 in PBST solution and 5% powdered milk. The presence of the antigen–antibody complex was revealed by the addition of a substrate solution containing 3,3,5,5-tetramethylbenzidine (Sigma Aldrich, St. Louis, MO, USA). The colorimetric reaction was stopped by adding 1 M sulfuric acid (H_2_SO_4_) per well and optical density (OD) measured at 492 nm using the ELX 800 plate reader (BioTek, Winooski, VT, USA). The optimal cutoff value of IgG ELISA was determined using a receiver operating characteristic (ROC) curve [16]. Thus, samples were considered positive when the OD was >0.348.

### 2.4. Molecular Analysis of LipL32-PCR

DNA from urine samples was extracted using the commercial Wizard SV Genomic DNA Purification System kit following the manufacturer’s instructions (Promega, Madison, USA). PCR assay targeted *lipL*32 gene (present majorly in pathogenic leptospires) using the primers specific for pathogenic leptospires LipL32-45F (5′-AAG CAT TAC CGC TTG TGG TG-3′) and LipL32-286R (5′-GAA CTC CCA CAG CGA TT-3) in a final volume of 25 µL following the conditions described in [15]. For each sample set, ultrapure water was used as a negative control, while 10 fg of DNA extracted from *Leptospira interrogans* serotype Copenhageni strain Fiocruz L1-130 was used as a positive control. All reactions occurred in the Gene Amp PCRSystem 9700 thermal cyclers (Applied Biosystems, Foster City, CA, USA). The total volume of each sample was analyzed using agarose gel electrophoresis (2%), stained with gel red and the DNA bands were visualized under ultraviolet light. The expected size of the amplicon was around 240 bp, varying slightly between the different species of *Leptospira* [17].

### 2.5. Statistical Analysis

The data were processed using the SPSS Statistics 20 software (IBM, Armonk, NY, USA). For quantitative variables, such as the optical density of the ELISA results in the different days/groups of the experiment, a descriptive analysis was employed. In addition, the Shapiro–Wilk test was used to assess the normal distribution of the data. Titer data observed in the MAT was converted to log10, and treated by geometric mean, according to [18]. In addition, seronegative animals were excluded from calculations and analysis of means (ANOVA). This correction ensured that a log titer corresponded to a tested dilution (1 to 1:100, 2 to 1:200, and 4 to 1:400), in the case of MAT. Nominal (non-parametric) variables, such as seroreactivity rate, serogroup dynamics, and PCR positivity were treated by McNemar tests (two samples) and Cochran test (Q test) (K samples). The paired Wilcoxon test was applied to determine equality in the paired serology. In addition, Fisher’s exact test was used to analyze 2 × 2 contingency tables at each time of the study for serological data. All analyses were performed at a 95% confidence level [19].

## 3. Results

All animals showed to be asymptomatic in all clinical examinations throughout the 365 days of the study. All the clinical exams were conducted by the same veterinarian.

### 3.1. Serology

Regarding MAT, a total of 69/94 (73.4%) vaccinated and 9/24 (37.5%) non-vaccinated dogs presented seroreactive at least once (*p* < 0.05; OR: 2.6), mainly against serogroups Icterohaemorrhagiae and Canicola. In Group A1 24/32 (75%) seroconverted, a result not different (*p* > 0.5) from Groups A2 (25/32—78.1%) and A3 (20/30—66.6%). Vaccinated dogs seroconverted majorly after D15 (*p* < 0.05). Vaccine #1 elicited a faster response, with a clear peak on D15 and a quick decrease (*p* < 0.05). For the others, the first peak was weaker, but other peaks occurred throughout the study. For all brands, seroreactivity was significantly higher than the control group until D180 (Figure 1). The serovars detected in the serology were Pomona, Icterohaemorrhagiae, Grippotyphosa, Hardjo, Canicola, and Copenhageni. In relation to ELISA results, all vaccine brands could successfully elicit an IgG response, while this difference was most clear at D30 for vaccine #1 (Figure 2).

### 3.2. Molecular Analysis

Considering the two main groups, in Group A (vaccinated), 44/94 (46.8%) were PCRpos at least once during the study, in contrast to 18/24 (75%) in Group B (unvaccinated) (*p* < 0.05; OR: 0.3 [0.1–0.8]). Of the 18 PCR positive dogs, 6 were seroreactive. Vaccinated animals presented 4/94 (4.2%) of multiple positivity in urinary PCR (more than one sampling), while for control group that rate was 4/24 (16.7%) (*p* < 0.05). The results demonstrate that vaccination could significantly reduce renal colonization by leptospires. Considering the three vaccinated subgroups, A1 presented 14/32 (43.7%) PCRpos, A2 16/32 (50%), and A3 14/30 (46.7%). There was no significant difference among the three vaccinated subgroups at any moment of the study. Temporal description of PCR results can be seen in Figure 3.

## 4. Discussion

In order to prevent leptospirosis in dogs, vaccination is essential, and is considered the frontline defense against the disease. Although it is well known that vaccination protects the animal against the acute form of the disease, it remains unclear the extent of the possible colonization prevention [20]. Furthermore, evidence on the protection elicited by serovars Canicola [11,21], Grippotyphosa [4,13,22], and Australis [23] in commercial vaccines is scarce. The duration of protection against these serovars is also inconsistent, as few animals in the control group showed clinical signs after experimental inoculation in the studies [4,13,22].

Another important factor is that studies on the effectiveness of vaccines generally use experimental and controlled groups in their methodologies regarding infection, feeding, and management. However, it is believed that these experimental studies may not represent the real conditions of the natural challenge that dogs suffer in situ [24]. In addition, the challenge with pathogenic *Leptospira* varies for in vivo testing, even using the same serovar [25]. Aiming at new scientific methodologies, the importance of considering the heterogeneity between studies, the best representation of the practical application of the treatment, and that experimental studies often distance the experimental sample from the reality of the population as a whole. We conducted this long-term study under natural conditions in an endemic area. The unexpected increase in antibodies in the control group and the positivity on molecular analysis for leptospires in dogs belonging to the control group could be related to environmental exposure to the pathogen or natural infection. These occurrences could not be characterized in this study. In addition, the inability to perform a culture isolation test for leptospires from the urine of the tested dogs prevents the determination of the actual environmental contamination capacity of the animals that tested PCR positive (carriers), vaccinated or non-vaccinated. In fact, the possibility of the vaccine-type effect on reduction in urinary shedding of viable leptospires should be considered. Most studies that evaluated the effectiveness of vaccines against leptospirosis were conducted under experimental and controlled conditions [10,21,26], while few studies have evaluated the effectiveness of vaccination in dogs under conditions of natural exposure to the environment and to leptospires [27,28]; thus, the real effect of vaccination against kidney colonization remains controversial. Nevertheless, it is important to remark that the exposure to infected dogs represents a risk for new infections and the establishment of renal carriers. It occurs in both endemic and non-endemic scenarios. Nevertheless, the transmission rate was not determined in this study. Noteworthy, the majority of studies focusing on the transmission of canine leptospirosis reinforces the role of rats as reservoirs and the contaminated environment, as well as pluviometry conditions [1].

The main objective when vaccinating dogs includes a longer and effective humoral response duration, what has been described from a range of 4–12 months [23,29,30]. The disadvantage of current bacterial vaccines arises from the fact that the induced immunity is directed mainly against the leptospiral lipopolysaccharide (LPS), a T-independent antigen, and therefore involves IgM antibodies and a lack of a memory response [31]. In addition, several studies have reported low antibody responses after vaccination with inactivated *Leptospira* vaccines [11,26,32]. Recently, a study has provided important results for new experiments, demonstrating the protective immunity induced with a subset of virulence-modifying proteins (VM) from *L. interrogans* against challenge infection, preventing the clinical pathogenesis of leptospirosis and leading to a marked reduction in leptospirosis target organ infection [33].

The difference observed between the MAT and ELISA results is related to the principles of each method. While MAT mainly detects agglutinating immunoglobulins (primarily IgM), our ELISA has been standardized for detection of IgG. The obtained results may be justified by the dynamics of the humoral response, with a first antibody curve, mainly by IgM, followed by an increase in IgG titers. Therefore, the decline in IgM titers, although partially compensated by the rise in IgG titers, is probably responsible for the reduction in seroreactivity in MAT. It has been reported that vaccination prevents clinical illness but does not protect against infection and excretion of bacteria, especially when infection occurs more than six months after vaccination [34]. In addition, most previous vaccine-related studies used only culture to demonstrate urinary excretion. Nevertheless, in asymptomatic dogs, shedding of viable of leptospires may happen in lower numbers, making isolation of leptospires more difficult. Furthermore, leptospiral shedding is intermittent, decreasing the sensitivity of direct tests. In this context, we believe that the usage of PCR may have been important to demonstrate a more accurate number of shedders.

According to our outcomes, we observed that all the vaccines significantly elicit the production of anti-*Leptospira* antibodies; although dogs remained seroreactive for up to 365 days, the difference between vaccinated/not vaccinated animals was clearer in the first six months of the study. Noteworthy, it was observed that, although the study was conducted in an endemic region and the animals were naturally exposed, none of the dogs presented clinical disease throughout the year of the study. Other studies corroborate our results, showing that no clinical signs were reported in the animals of the control group [25]. It is also noteworthy that in our studies there was no significant difference between the three vaccines on any of the studied variables. Although it was not the aim of the study to compare them, it was interesting to notice that, despite their different compositions, the results were essentially the same. Dogs play an important role in the epidemiology of *Leptospira* infection as they can act as both incidental and maintenance hosts with or without clinical symptoms, shedding leptospires in their urine [35]. With the exception of *Leptospira interrogans* serovar Canicola, for which the dog represents the main maintenance host, dogs are assumed to be incidental hosts for the infecting serovar and, consequently, shedding is likely to be brief when compared to that of reservoir hosts such as rodents [36,37].

One of the most controversial aspects is the role of vaccination on preventing infection and renal colonization with consequent shedding of leptospires, especially to dogs with high risk of infection [20], as we have studied for a long period of time. This is of great relevance, as the potential of infected asymptomatic dogs to spread the bacterium must always be kept in mind [38]. In particular, since vaccinated dogs may continue to excrete leptospires in the urine, they constitute a risk of human infection [8]. In this context, the possible effect of vaccination on avoiding or, at least, reducing the kidney colonization, would be highly desirable, since it would reduce the zoonotic risk and the transmission of pathogens among animal populations [32]. Vaccination data and dog challenge studies for serovars Icterohaemorrhagiae and Canicola suggest that some vaccines may allow a chronic carrier state in dogs and that the level of protection may vary between vaccines [11,26]. Other studies have shown the absence or low proportion of animals with renal shedding after experimental inoculation with Australis and Grippotyphosa serovars found in commercial vaccines [22,23,30]; in this context, we herein attend the call recently published by Esteves et al. [25], who claimed that “further studies are needed to assess the ability of vaccines to prevent renal carrier status after infection”. In our context, Grippotyphosa and Pomona are not common serovars. Thus, the lack of significant difference between the vaccines was not surprising.

A recent review showed that commercially available vaccines against leptospirosis can provide an overall protection of 84% against clinical disease, and up to 88% against renal carrier state, what is a very controversial number. Despite that, these findings should be interpreted with caution, since few experiments have tested the individual component of each vaccine. Another issue is that most vaccine studies used not only different parameters and methodologies, but also a highly variable test period of 3–35 days, to assess renal carrier status, making the comparison among the papers difficult [25].

Furthermore, the prevalence in the elimination of leptospires in the urine of dogs is quite variable depending on several factors. The prevalence in Thailand was 4.4% (12/273) of dogs shedding leptospires [39], in Ireland 7.1% (37/525) [35], 8.2% (41/500) in the USA [40], 31.1% in Iran [41], and 19.8% of dogs in Brazil [38], and our study in question showed 46.8% (44/94) of dogs shedding leptospires. In a German study, 1.5% (3/200) of healthy dogs were shedding leptospires [42], and in Switzerland, the shedding prevalence was 0.2% (1/408) [43]. This low European prevalence may be explained not only by the area being non-endemic, but also by having a broader immunity induced by the continuous vaccination of the canine population [39]. In this context, continuous vaccination of a large part of the population seems to reduce the prevalence of shedders over time.

The most important outcome of our study was that, although vaccinated dogs can still act as carriers of leptospires, a significant reduction was observed in the vaccinated animals, indicating a partial protection against renal colonization and consequent shedding that, although far from the ideal, cannot be ignored. Importantly, to our knowledge, this is the longest study ever conducted, and it was also conducted under natural conditions in an endemic area, with high infective pressure. We could observe that unvaccinated dogs were more likely to become asymptomatic carriers than the vaccinated ones and shed leptospires in the urine for a longer period (consecutive positive results).

## 5. Conclusions

Our results demonstrate that the currently available commercial vaccines, when applied in field conditions on naturally exposed dogs from an endemic area, were successful in eliciting an IgG-based long-term humoral response. In addition, it is known that vaccines do not prevent kidney colonization but provide partial protection against it and significantly reduced the number of shedders of leptospires in the urine.

## Figures and Tables

**Figure 1 animals-12-01788-f001:**
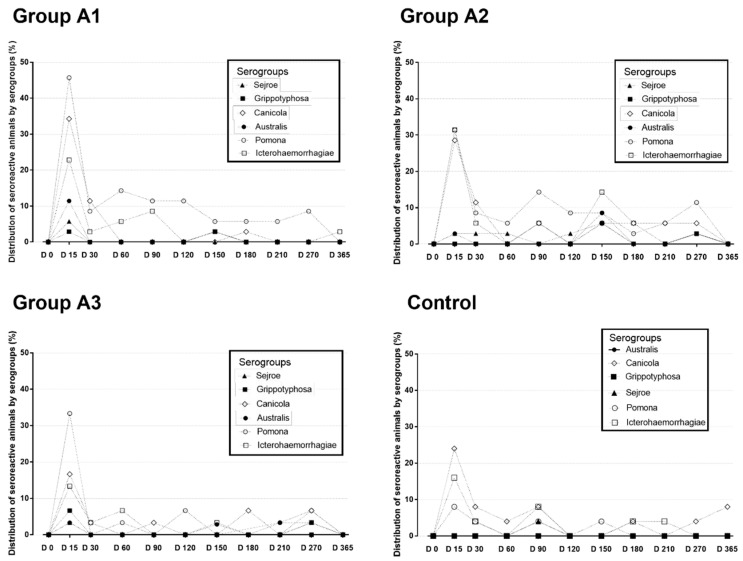
Distribution of seroreactive animals for MAT serogroups between the vaccinated (A1, A2, and A3) and unvaccinated (control) groups.

**Figure 2 animals-12-01788-f002:**
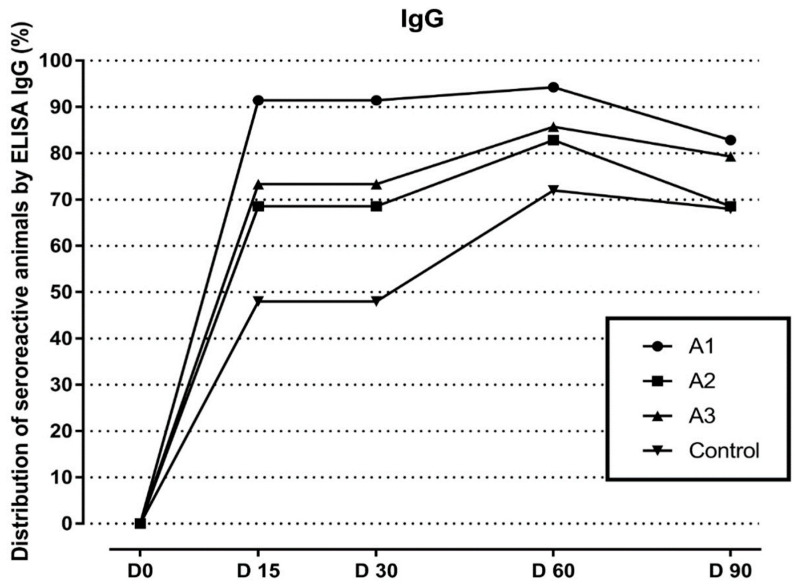
Distribution of seroreactive animals in the anti-Lipl32-IgG ELISA among vaccinated (A1, A2, and A3) and unvaccinated (control) groups.

**Figure 3 animals-12-01788-f003:**
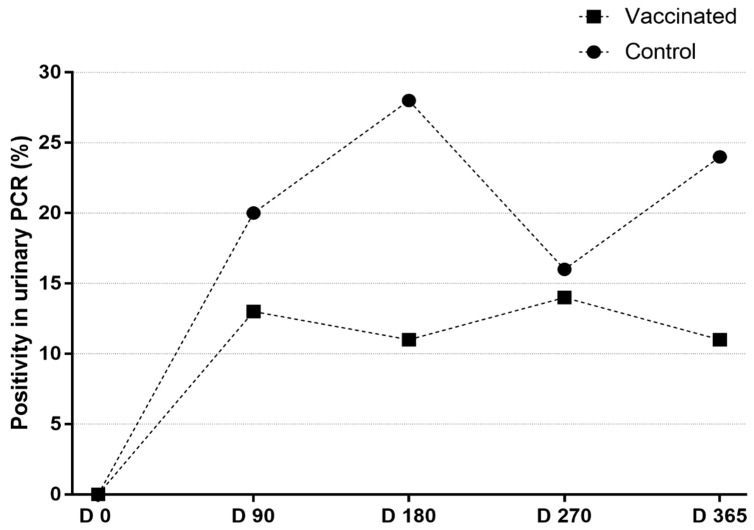
Temporal description of PCR results on vaccinated and unvaccinated groups of dogs.

## Data Availability

The data presented in this study are available from the corresponding author on request.

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
