# Peer review of "Effect of Vaccination against Leptospira on Shelter Asymptomatic Dogs Following a Long-Term Study"

_animals, 2022, doi:10.3390/ani12141788_

Round 1

Reviewer 1 Report

Dear colleagues

the manuscript describe the anibodies response to three different type of vaccinations against leptospiral infection in dogs (control group-unvaccinated dogs) and try to detect the correlation between the serological response with the possible chronic carrier status.

The manuscript needs a minor English language check (as reported below), and I would suggest to add a paraghaph reporting the "Study limitations" to increase the relevance of the study.

I have a question about the PCR molecular analysis: have you performed the sequencing on Positive urine samples? do yu find any correlation between the detectd ST and the vaccination? What serovars or serotypes did you detect? Why you did not perform the bacterial isolation culture (that would be usefull to better understand the effective viability of the Leptospires shedded by asymtptomatic infcted dogs). 

As previously reported, apparently  dogs have a low rate of viable Leptospire in urine shedding (back in memory 6-8%), and comparing to your results  vaccination is the most relatable choise for reducing urine shedding in infected dogs: can you discuss this point?

As previusly repoted, apparently the exposure to infected dogs, seems not representing a direct risk of infection (non endemic areaa) : can you discuss this point?

I would suggest the following considerations (line=number):

 Keyword3: I would change "silent"

45-47: Rephrase

50: Rephrase

58:Rephrase

67: add "leptospiral urine"

91-93: Rephrase

99: addd temperature

115; grammar check

110: in-house protocol?

105:space

146-147: Rephrase

209: remove capital letter

222: grammar check

235-237: grammar

224-227: Rephrase

236:"with consequent lepto shedding"

Author Response

Reviewer #1

Dear colleagues

Thank you very much for the time spent reviewing our manuscript. The suggestions were all very valuable and we are sure they will increase a lot the quality of our manuscript.

Q.1: The manuscript describes the antibodies response to three different type of vaccinations against leptospiral infection in dogs (control group-unvaccinated dogs) and try to detect the correlation between the serological response with the possible chronic carrier status.

The manuscript needs a minor English language check (as reported below), and I would suggest to add a paragraph reporting the "Study limitations" to increase the relevance of the study.

I have a question about the PCR molecular analysis: have you performed the sequencing on Positive urine samples?

R: Thank you for your comment. Actually, we have published preliminary results regarding sequencing of some of those amplicons (Santanna et al. 2021) before that long-term vaccination study was concluded. At that occasion, we have demonstrated that those dogs were 90.9% infected with Icterohaemorrhagiae strains (99.2-100% identity). It was not an unexpected result, since it is largely known that strains of Icterohaemorrhagiae, mainly Copenhageni L1-130, are the most frequent agents of canine and human leptospirosis in this region (Jaeger et al., 2018).

Q.2 Did you find any correlation between the detected ST and the vaccination?

R: No, we could not find that correlation.

Q.3 What serovars or serotypes did you detect?

R: At MAT we could detect antibodies against serogroups Pomona, Icterohaemorrhagiae, Grippotyphosa, Sejroe and Canicola. It was described in line 176 to 178.

Q.4 Why you did not perform the bacterial isolation culture (that would be useful to better understand the effective viability of the Leptospires shed by asymptomatic infected dogs). 

R: Indeed, it was originally one of the goals of the study. We have made an attempt of cultivating urine culture in the shelter, using an open flame. We have used EMJH liquid with antimicrobial cocktail STAFF and we immediately seeded the urine samples in the media. Unfortunately, despite the success of our field laboratory in other circumstances/studies, we could not recover isolates there. There was a lot of wind and we could not find an adequate place at the shelter. So, all the tubes have contaminated. We are evaluating a new attempt of urine culturing in the same shelter (and with some of the same animals of the study, if they are still there), using EMJH+STAFF and supplement with 1% rabbit serum, which is a proper media for primary isolation. The owner promised to let us use her own house this time (for a mini-lab), avoiding the wind and other adverse conditions.

Q5. As previously reported, apparently dogs have a low rate of viable Leptospires in urine shedding (back in memory 6-8%), and comparing to your results vaccination is the most relatable choice for reducing urine shedding in infected dogs: can you discuss this point?

R: Furthermore, the prevalence in the elimination of leptospires in the urine of dogs is quite variable depending on several factors, as discussed above. The prevalence in Thailand was 4.4% (12/273) of dogs shedding leptospires [38], in Ireland 7.1% (37/525) [34], 8.2% (41/500) in the USA [39], 31.1% in Iran [40], and 19.8% of dogs in Brazil [37], and our study in question showed 46.8% (44/94) of dogs shedding leptospires. In a German study, 1.5% (3/200) of healthy dogs were shedding leptospires [41], and in Switzerland, the shedding prevalence was 0.2% (1/408) [42]. This low European prevalence may be explained not only to the area being non-endemic, but also by having a broader immunity induced by the continuous vaccination of the canine population [38]. In this context, continuous vaccination of a large part of the population seems to reduce the prevalence of shedders along the years (Line 301 to 311).

Q6. As previously reported, apparently the exposure to infected dogs, seems not representing a direct risk of infection (non-endemic area): can you discuss this point?

R: We appreciate that observation. Indeed, that risk of infection was not assessed in our study. Statistical analysis did not include that data. As described in Introduction, the exposure to infected dogs represents a risk for new infections and the establishment of renal carrier. It occurs in both endemic and non-endemic scenarios. Nevertheless, the transmission rate was not determined in this study. Noteworthy that the majority of studies focusing in the transmission of canine leptospirosis reinforce the role of rats as reservoirs and the contaminated environment, as well as pluviometry conditions [1]. (Line 228 to 233).  

Q.6 I would suggest the following considerations (line=number): Keyword3: I would change "silent"; 45-47: Rephrase

R: Thanks for that point. In addition, leptospires affect the kidneys, being lodged for a long period, and may be excreted in the urine, remaining in the environment [6]. (Line 46-47)

50: Rephrase

  1. Thanks for that point. Dogs can be asymptomatic carriers, excreting serovars of different species in urine for long periods [8], and vaccination is referred to as the main tool for disease prevention [2]. (Line 50-51)

58: Rephrase

Phrase was altered.

67: add "leptospiral urine"

Added

91-93: Rephrase

Reformulated

99: add temperature

Added

115; grammar check

Rephrased

110: in-house protocol?

R: Thanks for that question. Yes, we standardized an in-house protocol for IgG Elisa. It was described in M&M, line 121

105: space

R: Done

146-147: Rephrase

Reformulated (Line 155 to 156)

209: remove capital letter

R: Done (Line 227)

222: grammar check

Phrase was altered

235-237: grammar

It was altered

224-227: Rephrase

Reformulated

236:"with consequent lepto shedding"

Added

Reviewer 2 Report

1. Some of the experimental results in the article are not presented in the form of figure. 

2. Whether it is reasonable that the control group's MAT results only showed a high percent on D15 in Fig.1, and repeated infections are bound to lead to increased antibody levels, but the trend in the control group is confusing.

3. The MAT titers shown in Fig.1 are more likely to indicate the problem. 

4. Materials and methods, the urine collection method is unclear, whether sterile and free of contamination.

5. line 122, 40mL?

6. line 125, what is the basis for OD value > 0.348? why?

7. line 129-130, are the primers universal?

8. Whether the veterinarian conducting the inspection is an official veterinarian and whether the inspection is standard?

9. Why was the first MAT test on day 15 after vaccination?

10. Did the dog use antibiotic drugs during the experimental cycle?

11. Is the dog with the first detection of Leptospira in urine still positive in the subsequent PCR test?

Author Response

Reviewer #2

Thank you very much for the time spent reviewing our manuscript. The suggestions were all very valuable and we are sure they will increase a lot the quality of our manuscript.

  1. Some of the experimental results in the article are not presented in the form of figure.

R: In fact, part of the results was not presented graphically. The authors prioritize graphics for harder results, as the textual part would be limited in terms of understanding.

  1. Whether it is reasonable that the control group's MAT results only showed a high percent on D15 in Fig.1, and repeated infections are bound to lead to increased antibody levels, but the trend in the control group is confusing. The MAT titers shown in Fig.1 are more likely to indicate the problem. 

R: It is important to notice that a great limitation of the study was the raise in antibodies and detection of leptospires in the control group, which represents an unexpected result. This raise may have happened due to natural infection by environmental conditions, but we could not determine this. Noteworthy to have in mind that long-term field studies have many variables that are difficult to control. (Line 218 to 223).

  1. Materials and methods, the urine collection method is unclear, whether sterile and free of contamination.

R: Urine samples were collected by catheterization by urethral tube. (Line 105).

  1. line 122, 40mL?

R: Done

  1. line 125, what is the basis for OD value > 0.348? why?

R: We appreciate that comment. Indeed, we have used an in-house Elisa and 0.348 was the cut-off used to better distinguish negative and positive animals, following ROC curve analysis. The description of “in-house” Elisa was included in the text (Line 132-134).

  1. line 129-130, are the primers universal?

R: The primers are specific for pathogenic leptospires, as informed (Line 138 to 139).

  1. Whether the veterinarian conducting the inspection is an official veterinarian and whether the inspection is standard?

R: Inspection was performed by the same veterinarian all the visitations. He is an experienced certified practitioner. He is not a member of official Public Health Department. (Line 88 to 91).

  1. Why was the first MAT test on day 15 after vaccination?
  2. R: We appreciate for that comment. Methodology was determined in order to assess the humoral response in the first 15 days, i.e. immediately after vaccination.

  1. Did the dog use antibiotic drugs during the experimental cycle?
  2. R: No, it is a very poor shelter and antibiotics and other drugs are very rarely available, depending on the charity of the associate members.

  1. Is the dog with the first detection of Leptospira in urine still positive in the subsequent PCR test?
  2. R: Sometimes, but not necessarily, since shedding is known to be intermittent, PCR results were very variable. (Lines 252-254)

Reviewer 3 Report

This study asks a basic question--how well do 3 different vaccines protect against leptospiral infection and the renal carrier state in dogs in an endemic area (and presumably high exposure pressure). The results are "shocking" in that almost 50% of the vaccinated dogs are PCR positive at some time during the 1-year study, as compared to 75% of controls. I feel the authors completely undersell the "most important outcome" of the study, which the authors state is the significant reduction in PCR positive tests in the vaccinates. That almost 50% of the vaccinates can shed leptospires at any time point, suggesting relative vaccine failure for preventing renal shedding (arguably the most important aspect of vaccination). Since none of the control dogs became sick in this 1-year period, despite a 75% PCR positive rate, vaccines showed no improved efficacy in this setting of preventing disease. 

General Comments:

1. The study is plagued with poor English throughout the paper.

2. It would have been great if the authors did MLST, MST or VNTR (or culture + PFGE) to try and determine the serovars (and strain) of the leptospires that were being shed by the vaccinated and control dogs. That would have added massive value to this study.

3. What are the prevalent leptospirosis serovars in this region of Brazil?

Specific comments:

Introduction

Line 53: The authors state that protection is dependent solely on humoral immunity, yet reference 28 would argue that cell-mediated immunity may be just as important.

Materials and Methods

Line 72: How many dogs were screened to find 118 that were seronegative? In an area that has endemic leptospirosis, I am surprised that any dog would be seronegative.

Line 74: How did they know these dogs were not vaccinated for at least 12 months? Dogs were kept in a shelter for the 12 months prior to the study?

Line 84: Were these dogs given a single vaccine, or was a booster given?

Line 99: I'm surprised the authors didn't do DNA isolation and then freeze the isolated DNA. Not even centrifugation prior to freezing? This doesn't seem like the most efficient method, although the authors still had remarkable success at isolating leptospiral DNA.

Figure 1 is a bit confusing, especially in context of Figure 2. One, why would the control dogs have a peak on D15? Two, In Figure 1, by D30 none of the dogs added up would exceed 40%, but in Figure 2 the IgG ELISA is up to 90% for at least 90 days. Are the authors implying that the IgG ELISA was more sensitive at detecting a persistent antibody response that the MAT? If so, this is an important point that needs to be more spelled out in the discussion.

Line 176--I've combined a few results in this comment, but in the controls, 18/24 were PCR positive but only 9 were seroreactive. That's remarkable, although not unexpected from other studies that show that seroreactivity does not equal the seropositive rate. I would be interested to know the PCR+ rate in the 9 dogs that were seroreactive.

Discussion

Line 211: As mentioned above, there are other studies that support that a robust cell-mediated immune response may be more important in protection from leptospirosis than a long humoral response.

Line 232-234: If Grippotyphosa and Pomona are not common serovars in the area, the lack of difference between the vaccines is not surprising. I would be cautious of the potential to infer that protection may be cross-serovar regardless of the vaccine. 

Line 235-237: Are the authors implying that all of these dogs were infected with the same "highly virulent strain"? The authors did not isolate or do molecular typing, so they have no idea if one or multiple serovars were infecting dogs. The fact that none of the controls became sick suggests this is not a highly virulent strain.

Line 239: Cultures would have been ideal in this study in an endemic area. It's unfortunate the authors did not incorporate that into the study.

Line 265: The authors state that the unvaccinated dogs were three times more likely to become asymptomatic carriers than the vaccinated ones. I don't see how that is calculated. 75% vs 47% does not equal a 3-to-1 ratio. I recognize that the Odds ratio they calculated was 0.3, but I also have a difficult time accepting that there is a real statistical difference between the PCR positive rate in the controls vs the vaccinates. I could throw in a number of variables that could alter these numbers: age of dogs (not presented in the study), the number of vaccines received prior to the 12-months free of vaccines, grouping within the shelter (e.g, maybe the control dogs were all housed in the "north wing" and the vaccinated dogs were housed in the "south wing", but rats had greater access and dogs more exposure in the "north wing"; or maybe housing the control dogs together just increased their general exposure rate).

As stated above, this study is a bit shocking in what I view as relative vaccine failure (especially given all previous studies which show a dramatic reduction in the risk of vaccinated dogs become renal shedders). The authors missed the opportunity to isolate/type the leptospires infecting these dogs, but they could still go back (maybe) and look at housing and age as variables in the positivity rate (instead of just "vaccinated vs control"). And I feel the authors are too generous in assigning any sort of protective benefit to vaccination. This is across the board failure to me.

Author Response

Reviewer #3

Thank you very much for the time spent reviewing our manuscript. The suggestions were all very valuable and we are sure they will increase a lot the quality of our manuscript.

This study asks a basic question--how well do 3 different vaccines protect against leptospiral infection and the renal carrier state in dogs in an endemic area (and presumably high exposure pressure). The results are "shocking" in that almost 50% of the vaccinated dogs are PCR positive at some time during the 1-year study, as compared to 75% of controls. I feel the authors completely undersell the "most important outcome" of the study, which the authors state is the significant reduction in PCR positive tests in the vaccinates. That almost 50% of the vaccinates can shed leptospires at any time point, suggesting relative vaccine failure for preventing renal shedding (arguably the most important aspect of vaccination). Since none of the control dogs became sick in this 1-year period, despite a 75% PCR positive rate, vaccines showed no improved efficacy in this setting of preventing disease. 

  1. Actually, it is clear that vaccination is far from being totally protective against renal colonization. Nevertheless, we believe it is more a question of expectative. Actually, manufacturers do not promise bacteriological protection for the current bacterin vaccines. Despite this, we cannot ignore the significant difference between the groups that was observed in that study. Some level of effect was observed and, considering the expectative of no protection, we decided that it cannot be ignored. So, a partial protection was indeed observed.

General Comments:

  1. The study is plagued with poor English throughout the paper.
  2. It would have been great if the authors did MLST, MST or VNTR (or culture + PFGE) to try and determine the serovars (and strain) of the leptospires that were being shed by the vaccinated and control dogs. That would have added massive value to this study.

R: Indeed, culturing was originally one of the goals of the study. We have made an attempt of cultivating urine culture in the shelter, using an open flame. We have used EMJH liquid with antimicrobial cocktail STAFF and we immediately seeded the urine samples in the media. Unfortunately, despite the success of our field laboratory in other circumstances/studies, we could not recover isolates there. There was a lot of wind and we could not find an adequate place at the shelter. So, all the tubes have contaminated.

Regarding molecular characterization of amplicons, we have published preliminary results regarding sequencing of some of those amplicons (Santanna et al. 2021) before that long-term vaccination study was concluded. At that occasion, we have demonstrated that dogs from that shelter were 90.9% infected with Icterohaemorrhagiae strains (99.2-100% identity). It was not an unexpected result, since it is largely known that strains of Icterohaemorrhagiae, mainly Copenhageni L1-130, are the most frequent agents of canine and human leptospirosis in this region (Jaeger et al., 2018).

  1. What are the prevalent leptospirosis serovars in this region of Brazil?

R: It is largely known that strains of Icterohaemorrhagiae, mainly Copenhageni L1-130, are the most frequent agents of canine and human leptospirosis in this region (JAEGER et al., 2018). (Line 59 to 61).

Specific comments:

Introduction

Line 53: The authors state that protection is dependent solely on humoral immunity, yet reference 28 would argue that cell-mediated immunity may be just as important.

R: Thanks for that question. Protection against leptospirosis through inactivated vaccines relies mainly predominantly on humoral immunity and strongly restricted to homologous serovar [9], directed towards Leptospira lipopolysaccharide (LPS) [10]. Despite that, cell-mediated immunity may be also important in this immunity. This was added to the text (Line 53 – 57)

Materials and Methods

Line 80: How many dogs were screened to find 118 that were seronegative? In an area that has endemic leptospirosis, I am surprised that any dog would be seronegative.

R: Thanks for that question. Actually, this number is quite uncertain, since serology (MAT) is conducted as a routine in this shelter, and seronegative dogs were separated to collect urine for PCR and integrate the study.

Line 74: How did they know these dogs were not vaccinated for at least 12 months? Dogs were kept in a shelter for the 12 months prior to the study?

R: The veterinarian who accompanied the dogs throughout the study has been the responsible (voluntarily) for the shelter for at least 15 years. (Line 88-91). Only rabies vaccines (which are for free, provided by the government) are regularly applied.

Line 84: Were these dogs given a single vaccine, or was a booster given?

R: Group A dogs were vaccinated with only one dose of vaccine, no booster. (Line 85-86)

Line 99: I'm surprised the authors didn't do DNA isolation and then freeze the isolated DNA. Not even centrifugation prior to freezing? This doesn't seem like the most efficient method, although the authors still had remarkable success at isolating leptospiral DNA.

  1. Regarding molecular characterization of amplicons, we have published preliminary results regarding sequencing of some of those amplicons (Santanna et al. 2021) before that long-term vaccination study was concluded. At that occasion, we have demonstrated that dogs of that shelter were 90.9% infected with Icterohaemorrhagiae strains (99.2-100% identity). It was not an unexpected result, since it is largely known that strains of Icterohaemorrhagiae, mainly Copenhageni L1-130, are the most frequent agents of canine and human leptospirosis in this region (Jaeger et al., 2018).

Figure 1 is a bit confusing, especially in context of Figure 2. One, why would the control dogs have a peak on D15? Two, In Figure 1, by D30 none of the dogs added up would exceed 40%, but in Figure 2 the IgG ELISA is up to 90% for at least 90 days. Are the authors implying that the IgG ELISA was more sensitive at detecting a persistent antibody response that the MAT? If so, this is an important point that needs to be more spelled out in the discussion.

R: That comment is imperative to us. The difference observed between the MAT and ELISA results is related to the principles of each method. While MAT mainly detects agglutinating immunoglobulins (primarily IgM), our ELISA has been standardized for detection of IgG. Considering the dynamics of the humoral response, with a first antibody curve, led by IgM, followed by an increase in IgG titers, the observed results are justified. Thus, the reduction in the seroreactivity of the animals throughout the study on the MAT is probably related to the fall in IgM titers, a fact that was offset by the increase in IgG titers. (Line 240 to 246).

Line 176--I've combined a few results in this comment, but in the controls, 18/24 were PCR positive but only 9 were seroreactive. That's remarkable, although not unexpected from other studies that show that seroreactivity does not equal the seropositive rate. I would be interested to know the PCR+ rate in the 9 dogs that were seroreactive.

R: Of the 18 PCR positive dogs, six were seroreactive. (Line 190)

Discussion

Line 211: As mentioned above, there are other studies that support that a robust cell-mediated immune response may be more important in protection from leptospirosis than a long humoral response.

R: Thank you, that observation was included in the text (line 56)

Line 232-234: If Grippotyphosa and Pomona are not common serovars in the area, the lack of difference between the vaccines is not surprising. I would be cautious of the potential to infer that protection may be cross-serovar regardless of the vaccine. 

R: We agree with that comment and the text was altered in lines 288-291

Line 235-237: Are the authors implying that all of these dogs were infected with the same "highly virulent strain"? The authors did not isolate or do molecular typing, so they have no idea if one or multiple serovars were infecting dogs. The fact that none of the controls became sick suggests this is not a highly virulent strain.

  1. Regarding molecular characterization of amplicons, we have published preliminary results regarding sequencing of some of those amplicons (Santanna et al. 2021) before that long-term vaccination study was concluded. At that occasion, we have demonstrated that dogs of that shelter were 90.9% infected with Icterohaemorrhagiae strains (99.2-100% identity). It was not an unexpected result, since it is largely known that strains of Icterohaemorrhagiae, mainly Copenhageni L1-130, are the most frequent agents of canine and human leptospirosis in this region (Jaeger et al., 2018).

Line 239: Cultures would have been ideal in this study in an endemic area. It's unfortunate the authors did not incorporate that into the study.

R: Indeed, culturing was originally one of the goals of the study. We have made an attempt of cultivating urine culture in the shelter, using an open flame. We have used EMJH liquid with antimicrobial cocktail STAFF and we immediately seeded the urine samples in the media. Unfortunately, despite the success of our field laboratory in other circumstances/studies, we could not recover isolates there. There was a lot of wind and we could not find an adequate place at the shelter. So, all the tubes have contaminated.

Line 265: The authors state that the unvaccinated dogs were three times more likely to become asymptomatic carriers than the vaccinated ones. I don't see how that is calculated. 75% vs 47% does not equal a 3-to-1 ratio. I recognize that the Odds ratio they calculated was 0.3, but I also have a difficult time accepting that there is a real statistical difference between the PCR positive rate in the controls vs the vaccinates. I could throw in a number of variables that could alter these numbers: age of dogs (not presented in the study), the number of vaccines received prior to the 12-months free of vaccines, grouping within the shelter (e.g, maybe the control dogs were all housed in the "north wing" and the vaccinated dogs were housed in the "south wing", but rats had greater access and dogs more exposure in the "north wing"; or maybe housing the control dogs together just increased their general exposure rate).

R: We agree with that sentence. The last paragraph of Discussion was altered. (Lines 317-318)

As stated above, this study is a bit shocking in what I view as relative vaccine failure (especially given all previous studies which show a dramatic reduction in the risk of vaccinated dogs become renal shedders).

R: As I mentioned earlier, the majority of previous vaccine-related studies used only culture to demonstrate urinary excretion. Nevertheless, in asymptomatic dogs, shedding of viable of leptospires may happen in lower number, what makes isolation of leptospires more difficult. We believe that the usage of PCR may have been important to demonstrate the real number of shedders. (Line 250 to 254).

The authors missed the opportunity to isolate/type the leptospires infecting these dogs, but they could still go back (maybe) and look at housing and age as variables in the positivity rate (instead of just "vaccinated vs control").

R: Those two points (culture and molecular characterizations) have been explained before, regarding other questions.  

And I feel the authors are too generous in assigning any sort of protective benefit to vaccination. This is across the board failure to me.

  1. Actually, it is clear that vaccination is far from being totally protective against renal colonization. Nevertheless, we believe it is more a question of expectative. Actually, manufacturers do not promise bacteriological protection for the current bacterin vaccines. Despite this, we cannot ignore the significant difference between the groups that was observed in that study. Some level of effect was observed and, considering the expectative of no protection, we decided that it cannot be ignored. So, a partial protection was indeed observed.

Round 2

Reviewer 1 Report

Dear colleagues

I greatly appreciate your efforts in responding to me accurately and taking suggestions.

I believe that your work is very significant and as of today deserves only a few very small changes.

Let me make a few more suggestions and thank you for accepting my comments.

Lines 215-219: please reformulate in this way

"The unexpected increase in antibodies in the control group and the positivity on molecular analysis for leptospire in dogs belonging to the control group could be related to environmental exposure to the pathogen or natural infection. These occurrences could not be characterized in this study. In addition, the inability to perform a culture isolation test for leptospires from the urine of the tested dogs prevents the determination of the actual environmental contamination capacity of the animals that tested PCR positive (carriers), vaccinated or non-vaccinated. In fact, the possibility of the vaccine-type effect on reduction of urinary shedding of viable leptospire should be considered."

Suggested for citation brand new paper: https://doi.org/10.3389/fcimb.2022.926994

239-242: Grammar check. 

Author Response

To the Editorial board of Animals,

July, 06th 2022

Dear Editor,

Please find enclosed our manuscript entitled " Effect of vaccination against Leptospira on shelter asymptomatic dogs followed a long-term study”, by Ricardo SantAnna; Maria Isabel Di Azevedo, Borges, A.L.; Ana Luiza dos Santos Baptista Borges; Luíza Aymeé; Gabriel Martins and Walter Lilenbaum, after corrections, for consideration for publication in your prestigious journal.

Dear colleagues,

Thank you very much for taking the time to review our manuscript. The suggestions were all very valuable and we are sure they greatly increased the quality of our manuscript. We will be delighted to see you accepted for publication in this journal.

Yours sincerely,

Walter Lilenbaum

Full Professor of Veterinary Bacteriology

Universidade Federal Fluminense

Rio de Janeiro, Brazil

[email protected]

Reviewer 2 Report

All my concerns have been solved.

Author Response

(The authors gave the same response as above.)
